# Optimal Body Mass Index Cut-off Point for Predicting Colorectal Cancer Survival in an Asian Population: A National Health Information Database Analysis

**DOI:** 10.3390/cancers12040830

**Published:** 2020-03-30

**Authors:** Nan Song, Dan Huang, Doeun Jang, Min Jung Kim, Seung-Yong Jeong, Aesun Shin, Ji Won Park

**Affiliations:** 1Cancer Research Institute, Seoul National University, Seoul 03080, Korea; nan.song@stjude.org (N.S.); doeunj@snu.ac.kr (D.J.); minjungkim@snuh.org (M.J.K.); syjeong@snu.ac.kr (S.-Y.J.); 2Department of Epidemiology and Cancer Control, St. Jude Children’s Research Hospital, Memphis, TN 38105, USA; 3Department of Preventive Medicine, Seoul National University College of Medicine, Seoul 03080, Korea; huangdan@ncc.re.kr; 4Division of Cancer Control & Policy, National Cancer Control Institute, National Cancer Center, Goyang 10408, Korea; 5Department of Surgery, Seoul National University College of Medicine and Hospital, Seoul 03080, Korea

**Keywords:** colorectal cancer, survival, body mass index, cut-off point

## Abstract

The optimal body mass index (BMI) range for predicting survival in Asian colorectal cancer patients is unknown. We established the most appropriate cut-off point of BMI to predict better survival in Asian colorectal cancer patients using a two-stage approach. Two cohorts of colorectal cancer patients were included in this study: 5815 hospital-based development cohort and 54,043 nationwide validation cohort. To determine the optimal BMI cut-off point at diagnosis, the method of Contal and O’Quigley was used. We evaluated the association between BMI and overall survival (OS) using the Cox proportional hazard model. During a median follow-up of 5.7 and 5.1 years for the development and the validation cohort, 1180 (20.3%) and 10,244 (19.0%) deaths occurred, respectively. The optimal cut-off of BMI identified as 20.2 kg/m^2^ (*p*_log-rank_ < 8.0 × 10^−16^) for differentiating between poorer and better OS in the development cohort. When compared to the patients with a BMI < 20.2 kg/m^2^, the patients with a BMI ≥ 20.2 kg/m^2^ had a significantly better OS (HR = 0.62, 95% CI = 0.54–0.72, *p* = 1.1 × 10^−10^). The association was validated in the nationwide cohort, showing better OS in patients with a BMI ≥ 20.2 kg/m^2^ (HR = 0.64, 95% CI = 0.60–0.67, *p* < 0.01). We suggest the use of a BMI value of 20.2 kg/m^2^ to predict survival in Asian colorectal cancer patients.

## 1. Introduction

Overweight or obesity defined by elevated body mass index (BMI) is considered an established risk factor for colorectal cancer [1]. This finding is supported by biological mechanisms that are influenced by obesity-induced insulin and growth factors, obesity-associated chronic inflammation, altered levels of adipocyte-secreted adiponectin and leptin, and sex hormones associated with cancer development [2]. However, the association between BMI and colorectal cancer mortality has been revealed to exhibit a U- or J-shaped relationship, which is referred to as the obesity paradox [3]. Prospective cohort studies and meta-analyses on colorectal cancer patients have reported that all-cause mortality was significantly increased in underweight and obese patients and decreased in overweight patients [4,5,6,7]. Although previous studies might have selection bias derived from healthy survivors, reverse causality, and collider stratification bias due to unmeasured factors, a retrospective cohort study with prospectively collected data using causal diagrams, which minimized potential methodological limitations, also reported that overweight patients had the best prognosis regardless of at-diagnosis or postdiagnosis BMI [8]. Furthermore, several Asian studies on the association between BMI and colorectal cancer prognosis have similarly observed that overweight patients, specifically with BMI ≥ 23 vs. <23 kg/m^2^ or with BMI 25–30 vs. 18.5–25 kg/m^2^, tended to show a favorable prognosis [9,10]. 

In a previous dose-response analysis of BMI at diagnosis and colorectal cancer prognosis, the highest survival rate was observed in patients with a BMI of 25–33 kg/m^2^, suggesting that normal weight defined by BMI 18.5–25 kg/m^2^ might not be optimal for survival [11]. However, Asians generally have BMIs in the lower range and lower BMI cut-off points for normal, overweight, and obesity defined by the World Health Organization (WHO) than those of Europeans [12]. Thus, the different optimal BMI for survival in colorectal cancer patients, compared to that of Europeans, could be suitably suggested, but there is no recommendation for a clear BMI cut-off point for predicting better survival in Asian colorectal cancer patients. Therefore, we derived the most appropriate cut-off point of BMI for better overall survival in a hospital-based colorectal cancer patient cohort and conducted validation using a nationwide health insurance database cohort. 

## 2. Results

### 2.1. Characteristics of Colorectal Cancer Patients and Associations with Overall Survival

Table 1 shows the general and clinical characteristics of colorectal cancer patients and their associations with overall survival in the development (SNUH) and validation (NHID) cohorts. Among 5815 colorectal cancer patients in the Seoul National University Hospital (SNUH) cohort, a total of 1180 (20.3%) patients died during a median follow-up of 5.7 years (range: 0.1–15.3 years, person-time: 36,912.0 years). Among the characteristics, age at diagnosis, BMI at diagnosis, sex, smoking status, hypertension, tumor site, tumor-node-metastasis (TNM) stage, perioperative chemotherapy, and perioperative radiotherapy were associated with overall survival (*p* < 0.05). Among 54,043 colorectal cancer patients from Korean National Health Information Database (NHID), a total of 10,244 (19.0%) patients died during a median follow-up of 5.1 years (range: 0.1–14.5 years, person-time: 300,395.8 years). There were statistically significant associations of age at diagnosis, sex, alcohol consumption, smoking, diabetes mellitus, hypertension, perioperative chemotherapy, and preoperative radiotherapy with overall survival (*p* < 0.05). 

### 2.2. Distribution of BMI at Diagnosis and Relationship with Overall Survival

The mean BMI at diagnosis was 23.3 kg/m^2^ (standard deviation [SD] = 3.1) in patients in the SNUH cohort and 23.9 kg/m^2^ (SD = 3.1) in patients from the NHID cohort. Appendix A shows the distribution of BMI at diagnosis in the development cohort. Appendix A shows a restricted cubic spline plot of the relationship between BMI at diagnosis and mortality in the development cohort. When BMIs at diagnosis that were lower than the 1^st^ (16.3 kg/m^2^) or higher than the 99^th^ (31.2 kg/m^2^) percentiles were considered potential outliers, a reverse J-shaped relationship between BMI and mortality was observed. A BMI > 30 kg/m^2^ showed no significant effects on mortality with a wide range of confidence intervals (95% CIs) of the hazard ratios (HRs).

### 2.3. Determination of the Optimal BMI Cut-Off Point for Overall Survival

Figure 1 shows absolute log-rank statistics to determine the optimal BMI cut-off point for overall survival by Contal and O’Quigley’s method in the SNUH cohort. The optimal BMI cut-off point at diagnosis was determined to be 20.2 kg/m^2^, which was the most significant cut-off point to differentiate patients with poorer and better survival (log-rank statistic = 96.69, Q-statistic = 3.07, *p* = 7.99 × 10^−16^). Each log-rank statistic according to the potential BMI cut-off point for overall survival is presented in Appendix A. The highest Youden index J was also observed for a BMI at diagnosis of 20.2 kg/m^2^ for overall survival. 

### 2.4. Characteristics and Survival Curves of Colorectal Cancer Patients by the Optimal BMI Cut-Off Point 

When colorectal cancer patients were divided by the selected BMI cut-off point (20.2 kg/m^2^), alcohol consumption, history of diabetes mellitus or hypertension, tumor site, TNM stage, and perioperative chemotherapy were associated with dichotomized BMI at diagnosis in the SNUH cohort (*p* < 0.05, Appendix A). In the NHID cohort, age at diagnosis, sex, alcohol consumption, smoking, diabetes mellitus, hypertension, and tumor site were associated with dichotomized BMI (*p* < 0.05, Appendix A). The Kaplan–Meier survival curves showed a clear difference in overall survival between patients with a BMI less than 20.2 kg/m^2^ and patients with a BMI equal to or greater than 20.2 kg/m^2^ in both the development and validation cohorts, as presented in Figure 2 (*p* for development cohort = 7.99 × 10^−16^, *p* for validation cohort = 6.10 × 10^−79^). The estimated 5-year OS was 86.5% for individuals with a BMI ≥ 20.2 kg/m^2^ and 70.9% for individuals with a BMI < 20.2 kg/m^2^ in the SNUH cohort and 83.9% and 74.6%, respectively, in the NHID cohort (Figure 2).

### 2.5. Multivariable Analysis of the Association between the Optimal BMI Cut-Off Point and Overall Survival

Multivariable analysis of the association between the BMI cut-off point and overall survival considering covariates are shown in Table 2. Colorectal cancer patients with a BMI ≥ 20.2 kg/m^2^ had better survival than patients with a BMI < 20.2 kg/m^2^ in the simple model (HR = 0.61, 95% CI = 0.53–0.70, *p* = 1.6 × 10^−12^) and in the multivariable model (HR = 0.62, 95% CI = 0.54–0.72, *p* = 1.1 × 10^−10^, Table 2). We also tested associations adjusted for all possible clinical characteristics and found a bit attenuated but statistically significant association with the same direction (HR = 0.66, 95% CI = 0.57–0.76, *p* = 1.8 × 10^−8^, Appendix A). These results were successfully validated in the NHID, showing better overall survival in patients with a BMI ≥ 20.2 kg/m^2^ recorded within at least 6 months prior to surgery (HR_simple model_ = 0.64, 95% CI = 0.56–0.63, *p* < 0.001, HR_multivariable model_ = 0.63, 95% CI = 0.60–0.67, *p* < 0.001, Table 2). Additional sensitivity analysis among patients with a BMI recorded at least within 3 months prior to surgery showed similar results (HR_simple model_ = 0.60, 95% CI = 0.56–0.64, *p* < 0.001, HR_multivariable model_ = 0.61, 95% CI = 0.57–0.65, *p* < 0.001, Table 2).

## 3. Discussion

In this study, we identified the outcome-oriented optimal BMI cut-off value and investigated the association with overall colorectal cancer survival using a hospital-based development cohort and a Korean nationwide validation cohort. This is the first study to determine the optimal BMI cut-off for colorectal cancer survival with a relatively large sample size and external validation using a nationwide database, suggesting that a BMI of at least 20.2 kg/m^2^ indicates better survival in Asian colorectal cancer patients.

Previous studies have observed a U-shaped relationship between BMI and colorectal cancer mortality [3,8,10,13] reporting the overweight with better survival and the elevated mortality in underweight patients. This obesity paradox was explained by a previous study that directly measured body composition and observed the adverse effect of low muscle mass or sarcopenia on the survival of colorectal cancer patients [13]. This is supported by the biological mechanism of muscle wasting caused by endotoxemia and the release of transforming growth factor-beta (TGF-β), which is especially promoted in gastrointestinal tract tumors and results in the majority of cancer deaths [14]. Experimentally, not only cancer but also cancer therapy activated signaling pathways leading to muscle wasting [15]. Moreover, colorectal cancer patients treated with surgical resection are likely to be subject to skeletal muscle loss due to acute inactivity caused by bed rest and/or chronically reduced activity. Since patients who are underweight are sensitive to muscle loss and sarcopenia and tend to have delayed recovery from surgery and longer hospital stays resulting in inactivity-mediated muscle loss, the increased mortality could be influenced by these negative health consequences [16].

In the obesity-survival paradox, whether the association reflects causality has been controversial in terms of several methodological issues, including uncontrolled confounding, selection bias derived from healthy survivors, and reverse causality [3,8]. In this analysis, we compared age- and sex-adjusted models and multivariate-adjusted models, including additional covariates (hypertension, tumor site, and TNM stage), which were related to BMI and overall survival, and found a similar result with statistical significance. Given that these results were robust as well, the association is unlikely to be influenced by other uncontrolled or unmeasured confounders. Furthermore, potential sampling selection bias was addressed through nationwide validation. In this study, colorectal cancer patients with a BMI equal to or greater than 20.2 kg/m^2^ at diagnosis had a better survival rate, and this finding was also validated with a strong significance in patients whose BMI was measured within 3 or 6 months before diagnosis, suggesting that this result was unlikely to be affected by reverse causality.

We chose only one cut-off point of BMI to dichotomize patients with better or worse survival. The reason was that none of the other possible cut-off points of BMI were observed with sufficient statistical significance to differentiate patients with poorer and better survival. When we conducted a preliminary association test between the WHO BMI categories and overall survival in this study population, we observed that underweight patients had significantly poorer survival, but overweight and obese patients had better survival than patients with normal weight (Appendix A). There was no significant association between morbid obesity and survival with a wide range of 95% CIs (Appendix A). This could be due to the small sample size of morbidly obese patients. However, if we consider that morbid obesity is not common in the Asian population [17], it might be more appropriate to focus on the minimum BMI suggested for better survival, especially in Asian colorectal cancer patients.

In terms of limitations of this study, first, TNM stage could not be adjusted in the analysis of the validation cohort because the NHID did not provide information on TNM stage. Instead, the multivariable-adjusted model additionally included chemotherapy as a covariate. However, given that the prevalence of colorectal cancer patients with a BMI < 20.2 kg/m^2^ was relatively higher in the SNUH (15.7%) than in the KNHID (10.1%), the more advanced cancer patients might be included in the SNUH. Second, because BMI is a suboptimal approximation of body adiposity, further studies on optimal cut-off points of various body size indicators, such as waist circumference [18], waist-to-hip ratio [19], ideal weight ratio [20], and body surface area [20], would improve the validity of the results.

Nevertheless, this study has several strengths. One major strength is that this is the first study to develop and validate the optimal BMI cut-off point for better survival in Asian colorectal cancer patients. Furthermore, we included a relatively large sample of representative colorectal cancer patients in Korea. Because the SNUH is the major national hospital located in the capital city of Korea as well as one of the largest tertiary care hospitals that provides the greatest number of colorectal cancer operations according to the Health Insurance Review & Assessment Service in Korea [21], cancer patients from across the nation would visit and undergo surgical treatment. The representativeness in the SNUH cohort was also supported by the fact that our findings were validated in the nationwide cohort with strong statistical significance. International validation in East Asian countries would increase the generalizability.

## 4. Materials and Methods

### 4.1. Study Population

Two cohorts of colorectal cancer patients were included in this study (Figure 3). In a retrospective cohort study with prospective follow-up, colorectal cancer patients who underwent surgical resection were recruited from SNUH, as described elsewhere [22]. A total of 5874 colorectal cancer patients were enrolled in the SNUH cohort between January 2002 and December 2014. We excluded 59 patients who died within 30 days after colorectal surgery to minimize effects directly related to surgery. A total of 5815 patients remained in the development cohort. This study was approved by the institutional review board (IRB) of the SNUH and the ethics committee approved this retrospective study with a waiver of written informed consent (IRB No. 1906-116-1041, 1908-094-1055).

The NHID provided by the National Health Insurance Service (NHIS) [23], which covers eligibility, national health screening, healthcare usage, long-term care insurance, and healthcare provider data of approximately 98% of the Korean population, was used for validation. Colorectal cancer patients were defined as individuals who were diagnosed with C18.0-20.0 as a main or sub-disease based on the 10th international Classification of Disease (ICD-10) and who underwent surgical treatment. Among 202,723 eligible colorectal cancer patients from 2002 to 2017, we excluded 39,499 patients who did not have BMI records. We also excluded 15,043 patients who died within 30 days after colorectal surgery or whose follow-up was less than 30 days. Among the remaining 148,203 patients, we only included 54,043 patients with information on BMI within 6 months (183 days) before the date of surgery in this analysis. The sensitivity analysis was additionally restricted to 41,159 patients whose information on BMI within 3 months (92 days) before the date of surgery was available.

### 4.2. Data Collection

In a developmental cohort study of SNUH, general demographic information (age at diagnosis and sex), lifestyle (alcohol consumption and smoking status), history of diabetes mellitus and hypertension, anthropometry at the time of admission (height and weight), pathological (tumor site and TNM stage) and treatment (perioperative chemotherapy) data were extracted from patients’ medical records. From the validation cohort, we extracted the same variables except for TNM stage, which was not provided by the NHID. BMI at diagnosis was considered as the main exposure in this analysis. BMI was calculated from the patient’s weight (kg) divided by square of height (m^2^) in a standard way. Since the BMI at the time of diagnosis of colorectal cancer was not available in the NHID claim data, we used the BMI closest to the date of surgery within 3 or 6 months before colorectal surgery from health check-up data. Death certificate data provided by Statistics Korea by Korean Statistical Information Service (KOSIS) [24] were individually linked to each colorectal cancer patient in the SNUH and NHID cohorts.

The patients were enrolled, underwent surgery in SNUH, and were followed up every 3 or 6 months for the first 2 years and every 6 months or 1 year for 3 years thereafter. For the analysis, patients from SNUH were followed up until death or 23 April, 2017. The death information of the validation cohorts from NHID were annually followed up until 18 July, 2017. Overall survival was defined as the time from the date of surgery to the date of death from any cause in patients who died. Surviving patients were censored at the date of the last follow-up.

### 4.3. Statistical Analysis

To assess the associations of patients’ characteristics with overall survival, we used the log-rank test for categorical variables and the linear-rank test for continuous variables. The distribution of BMI at diagnosis was visualized in a histogram and a box-plot. To visually depict the relationship between BMI at diagnosis and mortality, a restricted cubic spline Cox regression model was fit with 4 knots and a reference value of 20 kg/m^2^. To determine the optimal cut-off point of BMI at diagnosis for predicting survival, we used the method of Contal and O’Quigley [25] in the development cohort. Using this method, BMI, as a continuous variable, was systematically dichotomized based on log-rank statistics. The commonly used Youden index (J = sensitivity + specificity − 1) method [26] was also used but was considered inadequate for time-to-event analysis. We compared patient characteristics between the dichotomized BMI groups using the chi-square test for categorical variables and the T-test for continuous variables. We evaluated the association between dichotomized BMI at diagnosis and overall survival using the Cox proportional hazard model and calculated HRs and 95% CIs adjusted for age and sex in the simple model or in the multivariable model. The variables that were adjusted in the multivariable model were selected based on both their association with BMI and overall survival, missing values were less than 10% of the study population, and multicollinearity among clinical characteristics; the adjusted variable were as follows: age, sex, history of hypertension, tumor site, and TNM stage. For the validation cohort, the multivariable model additionally included chemotherapy as a covariate because TNM stage information was not provided by the NHID. Kaplan–Meier curves for overall survival were constructed to visualize the patient’s time to event according to BMI groups dichotomized by the optimal cut-off point. For all statistical analyses, a two-sided *p*-value of 0.05 was considered as the threshold for significance, and SAS 9.4 was used. Figures were produced by using STATA SE 14.

## 5. Conclusions

From this study, we suggest the use of a BMI cut-off of 20.2 kg/m^2^ to predict better survival in Asian colorectal cancer patients. This cut-off point could support health professionals in improving the survival of Asian colorectal cancer patients. Furthermore, the comprehensive nutritional assessment, counseling, and support for underweight for underweight patients are needed to improve their overall prognosis. Additional international studies across Asian countries are warranted to further establish the optimal BMI cut-off point for the survival of colorectal cancer patients.

## Figures and Tables

**Figure 1 cancers-12-00830-f001:**
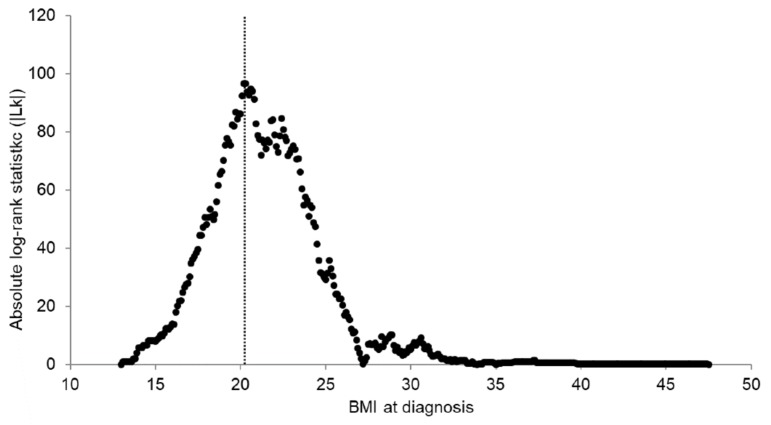
Determination of the optimal BMI cut-off point for overall survival using Contal and O’Quigley’s method in the SNUH cohort. The dashed line demarcates the optimal BMI cut-off point: 20.2 (log-rank statistic: 96.69, Q-statistic: 3.07, *p* <0.01).

**Figure 2 cancers-12-00830-f002:**
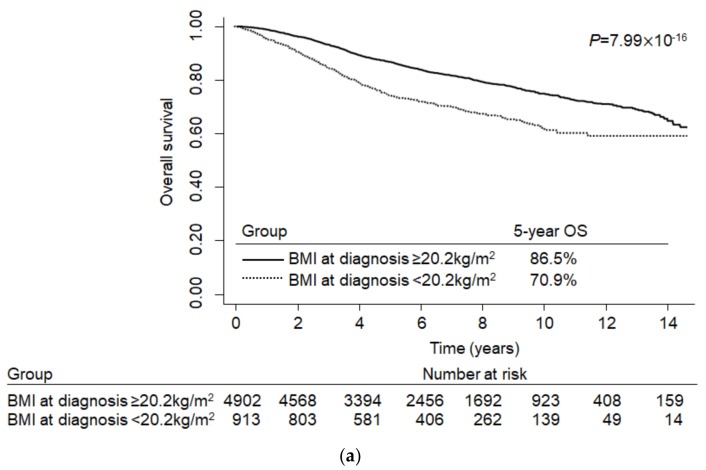
Kaplan–Meier curves for the optimal cut-off point of BMI at diagnosis and overall survival in colorectal cancer patients: (**a**) Development cohort (Seoul National University Hospital (SNUH)); (**b**) validation cohort (Korean National Health Information Database (NHID)).

**Figure 3 cancers-12-00830-f003:**
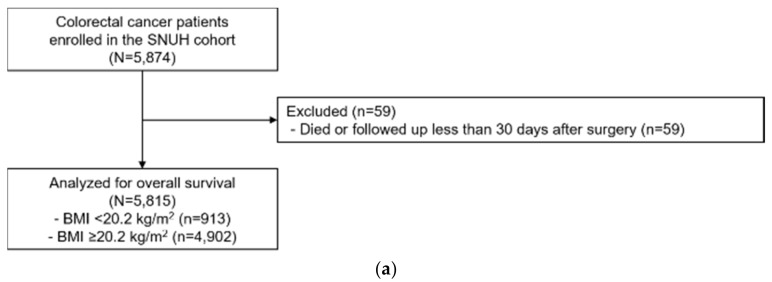
CONSORT diagram of colorectal cancer patients who underwent surgical resection: (**a**) Development cohort (SNUH); (**b**) Validation cohort (NHID).

**Table 1 cancers-12-00830-t001:** Characteristics of colorectal cancer patients and association with overall survival.

Characteristics	NHID								
	Event		*p* ^a^	All		Event		*p* ^a^
	(*N* = 1,180, 20.3%)			(*N* = 54,043)		(*N* = 10,244, 19.0%)		
	(%)	*N*	(%)		*N*	(%)	*N*	(%)	
Follow-up (years), median (range)	5.1 (0.1–14.5)								
Person-time (years)	300,395.8								
Age at diagnosis (years), mean (SD)	(11.2)	66.3	(11.8)	<0.01	62.7	(10.3)	66.1	(10.8)	<0.01
BMI at diagnosis (kg/m^2^), mean (SD)	(3.1)	22.8	(3.4)	<0.01	23.9	(3.1)			
Sex				<0.01					<0.01
Men	(61.5)	799	(67.7)		34,109	(63.1)	6916	(67.5)	
Women	(38.5)	381	(32.3)		19,934	(36.9)	3328	(32.5)	
Alcohol drinking status				0.20					0.02
Never	(67.8)	795	(67.4)		22,114	(40.9)	4459	(43.5)	
Ever	(30.6)	343	(29.1)		20,384	(37.7)	3862	(37.7)	
Unknown	(1.6)	42	(3.6)		11,545	(21.4)	1923	(18.8)	
Smoking status				<0.01					<0.01
Never	(78.9)	884	(74.9)		31,802	(58.9)	5944	(58.0)	
Ever	(19.5)	256	(21.7)		20,676	(38.3)	4002	(39.1)	
Unknown	(1.5)	40	(3.4)		1565	(2.9)	298	(2.9)	
Diabetes mellitus				0.09					<0.01
No	(85.6)	994	(84.2)		44,036	(81.5)	7817	(76.3)	
Yes	(14.3)	185	(15.7)		10,007	(18.5)	2427	(23.7)	
Unknown	(0.0)	1	(0.0)		-		-		
Hypertension				<0.01					<0.01
No	(64.3)	725	(61.4)		34,976	(64.7)	6198	(60.5)	
Yes	(35.7)	454	(38.5)		19,067	(35.3)	4046	(39.5)	
Unknown	(0.0)	1	(0.0)		-		-		
Tumor site				<0.01					0.06
Colon	(64.7)	487	(41.3)		38,685	(71.6)	6998	(68.3)	
Rectum	(35.3)	693	(58.7)		13,771	(25.5)	2750	(26.8)	
Unknown		-			1587	(2.9)	496	(4.8)	
TNM stage				<0.01					
I	(23.0)	148	(12.5)		-		-		
II	(36.3)	388	(32.9)		-		-		
III	(40.7)	644	(54.6)		-		-		
Perioperative chemotherapy				<0.01					<0.01
No	(29.2)	408	(34.6)		41,742	(77.2)	5867	(57.3)	
Yes	(59.8)	709	(60.1)		12,301	(22.8)	4377	(42.7)	
Unknown	(11.0)	63	(1.1)		-		-		
Perioperative radiotherapy				<0.01					<0.01
No	(68.7)	822	(69.7)		48,334	(89.4)	7891	(77.0)	
Yes	(19.5)	294	(24.9)		5709	(10.6)	2353	(23.0)	
Unknown	(11.8)	64	(5.4)		-		-		

**Table 2 cancers-12-00830-t002:** Association between BMI cut-off point and overall survival in colorectal cancer patients.

BMI	Total		Event		Age and Sex-Adjusted Model ^a^			Mutivariable-Adjusted Model ^b^		
	*N*	(%)	*N*	(%)	HR	(95% CI)	*p*	HR	(95% CI)	*p*
**Development**										
BMI at diagnosis (kg/m^2^), SNUH										
<20.2	913	(15.7)	265	(22.5)	1.00	(ref.)		1.00	(ref.)	
≥20.2	4902	(84.3)	915	(77.5)	0.61	(0.53–0.70)	1.6 × 10^−12^	0.62	(0.54–0.72)	1.1 × 10^−10^
**Validation**										
BMI recorded less than 6 months prior to surgery (kg/m^2^), NHID										
<20.2	5462	(10.1)	1514	(14.8)	1.00	(ref.)		1.00	(ref.)	
≥20.2	48,581	(89.9)	8730	(85.2)	0.64	(0.56–0.63)	<0.001	0.63	(0.60–0.67) ^c^	<0.001
BMI recorded less than 3 months prior to surgery (kg/m^2^), NHID										
<20.2	4112	(10.0)	1077	(15.2)	1.00	(ref.)		1.00	(ref.)	
≥20.2	37,047	(90.0)	6022	(84.8)	0.60	(0.56–0.64)	<0.001	0.61	(0.57–0.65) ^c^	<0.001

Abbreviation: BMI (body mass index), HR (hazard ratio), CI (confidence interval), SNUH (Seoul National University Hospital), and NHID (National Health Insurance Database). ^a^ Cox proportional hazard model adjusted for age and sex. ^b^ Cox proportional hazard model adjusted for age, sex, hypertension, tumor site, and TNM stage. ^c^ Cox proportional hazard model adjusted for age, sex, hypertension, and tumor site, and chemotherapy.

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
