# Peer review of "Optimal Body Mass Index Cut-off Point for Predicting Colorectal Cancer Survival in an Asian Population: A National Health Information Database Analysis"

_cancers, 2020, doi:10.3390/cancers12040830_

Round 1
Reviewer 1 Report
- There are many factors affect the survival of colorectal cancer. Such as tumor differentiation, primary tumor site (colon or rectum), surgical margin, etc. Why the authors don't include these common characteristics in the article?
- Why the authors exclude the stage IV CRC patients? Low BMI patients among the group seem to have poor prognosis.
- Did the authors have any suggestion in clinical practice according to the result?
- The subtitle should be arranged in the following order: Introduction, Materials and Methods, Result, Discussion, Conclusion.
Author Response
Reviewer #1
Comment 1: There are many factors affect the survival of colorectal cancer. Such as tumor differentiation, primary tumor site (colon or rectum), surgical margin, etc. Why the authors don't include these common characteristics in the article?
Response: The reviewer raises a relevant concern about the potential effect of the other clinical characteristics. We initially considered the more possible clinical characteristics. But the variables that were adjusted in the multivariable model were selected based on both their association with BMI and overall survival, and missing values were less than 10% of the study population. Furthermore, we considered multicollinearity among clinical characteristics, which could undermine the statistical significance. Based on this, the adjusted variable was selected as follows: age, sex, history of hypertension, tumor site, and TNM stage. For your information, we added the results adjusted for other clinical characteristics including age, sex, hypertension, tumor site, TNM stage, tumor grade, bowel obstruction, bowel perforation, and ASA grade in the Table S3.
[Lines 116-119]
We also tested associations adjusted for all possible clinical characteristics and found a bit attenuated but statistically significant association with the same direction (HR=0.66, 95% CI=0.57-0.76, p=1.8×10-8, Table S3).
[Lines 298-301]
The variables that were adjusted in the multivariable model were selected based on both their association with BMI and overall survival, missing values were less than 10% of the study population, and multicollinearity among clinical characteristics; the adjusted variable were as follows: age, sex, history of hypertension, tumor site, and TNM stage.
[Table S3]
|
Table S3. Association between BMI cut-off point and overall survival in SNUH colorectal cancer patients according to the different multivariable models |
|||||||||||||
|
|
Total |
Event |
Age and sex-adjusted modela |
Mutivariable-adjusted model 1b |
Mutivariable-adjusted model 2c |
||||||||
|
N |
(%) |
N |
(%) |
HR |
(95% CI) |
P |
HR |
(95% CI) |
P |
HR |
(95% CI) |
P |
|
|
BMI at diagnosis (kg/m2), SNUH |
|||||||||||||
|
<20.2 |
913 |
(15.7) |
265 |
(22.5) |
1.00 |
(ref.) |
1.00 |
(ref.) |
1.00 |
(ref.) |
|||
|
≥20.2 |
4,902 |
(84.3) |
915 |
(77.5) |
0.61 |
(0.53-0.70) |
1.6×10-12 |
0.62 |
(0.54-0.72) |
1.1×10-10 |
0.66 |
(0.57-0.76) |
1.8×10-8 |
|
Abbreviation: BMI (body mass index), HR (hazard ratio), CI (confidence interval), and SNUH (Seoul National University Hospital). |
|||||||||||||
|
aCox proportional hazard model adjusted for age and sex. |
|||||||||||||
|
bCox proportional hazard model adjusted for age, sex, hypertension, tumor site, and TNM stage. |
|||||||||||||
|
cCox proportional hazard model adjusted for age, sex, hypertension, tumor site, TNM stage, tumor grade, bowel obstruction, bowel perforation, and ASA grade. |
|||||||||||||
Reviewer 2 Report
This work is very interesting because it demonstrates the paradox of obesity in colorectal cancer survival. The identified cutoff of a BMI> 20 kg / m2 selects a group of patients (overweight) that has a higher survival than underweight (BMI <20) or obese (BMI> 30) patients. Both in underweight and obese patients there is the phenomenon of malnutrition leading to the development of sarcopenia, the latter is known to be a negative prognostic factor in cancer patients. In this study it would have been useful to make a further analysis with bioimpendentiometry to get a general overview of all the compartments of the body and to analyze the lean mass, muscle mass, fat mass and quantity of intra and extra cellular water. The loss of lean mass and consequently of the muscle mass that is observed in both underweight and obese subjects leads to sarcopenia, as in both cases there is at the base a state of chronic inflammation which activates a series of proinflammatory cytokines that could be responsible for lower survival in cancer patients. For these patients, nutritional support should be identified in order to improve BMI and avoid sarcopenia.
Author Response
Reviewer #2
Comment: This work is very interesting because it demonstrates the paradox of obesity in colorectal cancer survival. The identified cutoff of a BMI> 20 kg / m2 selects a group of patients (overweight) that has a higher survival than underweight (BMI <20) or obese (BMI> 30) patients. Both in underweight and obese patients there is the phenomenon of malnutrition leading to the development of sarcopenia, the latter is known to be a negative prognostic factor in cancer patients. In this study it would have been useful to make a further analysis with bioimpendentiometry to get a general overview of all the compartments of the body and to analyze the lean mass, muscle mass, fat mass and quantity of intra and extra cellular water. The loss of lean mass and consequently of the muscle mass that is observed in both underweight and obese subjects leads to sarcopenia, as in both cases there is at the base a state of chronic inflammation which activates a series of proinflammatory cytokines that could be responsible for lower survival in cancer patients. For these patients, nutritional support should be identified in order to improve BMI and avoid sarcopenia.
Response: We agree with the reviewer that the loss of lean mass and muscle mass could related to sarcopenia and the worse prognosis in colorectal cancer patients. Thus, we mentioned the relevant contents about the body composition in underweight patients and their prognosis in the discussion section, since we could not conduct the further analysis on body composition.
[Lines 134-145]
This obesity paradox was explained by a previous study that directly measured body composition and observed the adverse effect of low muscle mass or sarcopenia on the survival of colorectal cancer patients [13]. This is supported by the biological mechanism of muscle wasting caused by endotoxemia and the release of transforming growth factor-beta (TGF-β), which is especially promoted in gastrointestinal tract tumors and results in the majority of cancer deaths [15]. Experimentally, not only cancer but also cancer therapy activated signaling pathways leading to muscle wasting [16]. Moreover, colorectal cancer patients treated with surgical resection are likely to be subject to skeletal muscle loss due to acute inactivity caused by bed rest and/or chronically reduced activity. Since patients who are underweight are sensitive to muscle loss and sarcopenia and tend to have delayed recovery from surgery and longer hospital stays resulting in inactivity-mediated muscle loss, the increased mortality could be influenced by these negative health consequences [17].
Reviewer 3 Report
Comments for the author
Remarks to the author:
- There are many grammatical errors which needs to be improved.
- Please provides few references to Asian studies in the introduction section.
- Discussion section is very long and diffused in its current form, I would recommend rewriting this section in more concise manner.
Author Response
Reviewer #3
Comment 1: There are many grammatical errors which needs to be improved.
Response: This manuscript was edited by the qualified native English speaking editors and we added the editorial certificate.
Comment 2: Please provides few references to Asian studies in the introduction section.
Response: The reviewer suggested, we cited Asian studies and added the relevant sentence in the introduction section.
[Line 50-53]
Furthermore, several Asian studies on the association between BMI and colorectal cancer prognosis have similarly observed that overweight patients, specifically with BMI ≥23 vs. <23 kg/m2 or with BMI 25-30 vs. 18.5-25 kg/m2, tended to show a favorable prognosis [9,10].
Comment 3: Discussion section is very long and diffused in its current form, I would recommend rewriting this section in more concise manner.
Response: As the reviewer suggested, we shortened the discussion section and the several sentences and paragraphs were re-organized in context.
[Lines 126-239]
In this study, we identified the outcome-oriented optimal BMI cut-off value and investigated the association with overall colorectal cancer survival using a hospital-based development cohort and a Korean nationwide validation cohort. This is the first study to determine the optimal BMI cut-off for colorectal cancer survival with a relatively large sample size and external validation using a nationwide database, suggesting that a BMI of at least 20.2 kg/m2 indicates better survival in Asian colorectal cancer patients.
Previous studies have observed a U-shaped relationship between BMI and colorectal cancer mortality [8,10,13,14] reporting the overweight with better survival and the elevated mortality in underweight patients. This obesity paradox was explained by a previous study that directly measured body composition and observed the adverse effect of low muscle mass or sarcopenia on the survival of colorectal cancer patients [13]. This is supported by the biological mechanism of muscle wasting caused by endotoxemia and the release of transforming growth factor-beta (TGF-β), which is especially promoted in gastrointestinal tract tumors and results in the majority of cancer deaths [15]. Experimentally, not only cancer but also cancer therapy activated signaling pathways leading to muscle wasting [16]. Moreover, colorectal cancer patients treated with surgical resection are likely to be subject to skeletal muscle loss due to acute inactivity caused by bed rest and/or chronically reduced activity. Since patients who are underweight are sensitive to muscle loss and sarcopenia and tend to have delayed recovery from surgery and longer hospital stays resulting in inactivity-mediated muscle loss, the increased mortality could be influenced by these negative health consequences [17].
In the obesity-survival paradox, whether the association reflects causality has been controversial in terms of several methodological issues, including uncontrolled confounding, selection bias derived from healthy survivors, and reverse causality [3,8]. In this analysis, we compared age- and sex-adjusted models and multivariate-adjusted models, including additional covariates (hypertension, tumor site, and TNM stage), which were related to BMI and overall survival, and found a similar result with statistical significance. Given that these results were robust as well, the association is unlikely to be influenced by other uncontrolled or unmeasured confounders. Furthermore, potential sampling selection bias was addressed through nationwide validation. In this study, colorectal cancer patients with a BMI equal to or greater than 20.2 kg/m2 at diagnosis had a better survival rate, and this finding was also validated with a strong significance in patients whose BMI was measured within 3 or 6 months before diagnosis, suggesting that this result was unlikely to be affected by reverse causality.
We chose only one cut-off point of BMI to dichotomize patients with better or worse survival. The reason was that none of the other possible cut-off points of BMI were observed with sufficient statistical significance to differentiate patients with poorer and better survival. When we conducted a preliminary association test between the WHO BMI categories and overall survival in this study population, we observed that underweight patients had significantly poorer survival, but overweight and obese patients had better survival than patients with normal weight (Figure S2). There was no significant association between morbid obesity and survival with a wide range of 95% CIs (Figure S2). This could be due to the small sample size of morbidly obese patients. However, if we consider that morbid obesity is not common in the Asian population[18], it might be more appropriate to focus on the minimum BMI suggested for better survival, especially in Asian colorectal cancer patients.
In terms of limitations of this study, first, TNM stage could not be adjusted in the analysis of the validation cohort because the NHID did not provide information on TNM stage. Instead, the multivariable-adjusted model additionally included chemotherapy as a covariate. However, given that the prevalence of colorectal cancer patients with a BMI <20.2 kg/m2 was relatively higher in the SNUH (15.7%) than in the KNHIC (10.1%), the more advanced cancer patients might be included in the SNUH. Second, because BMI is a suboptimal approximation of body adiposity, further studies on optimal cut-off points of various body size indicators, such as waist circumference [19], waist-to-hip ratio [20], ideal weight ratio [21], and body surface area [21], would improve the validity of the results.
Nevertheless, this study has several strengths. One major strength is that this is the first study to develop and validate the optimal BMI cut-off point for better survival in Asian colorectal cancer patients. Furthermore, we included a relatively large sample of representative colorectal cancer patients in Korea. Because the SNUH is the major national hospital located in the capital city of Korea as well as one of the largest tertiary care hospitals that provides the greatest number of colorectal cancer operations according to the Health Insurance Review & Assessment Service in Korea [22], cancer patients from across the nation would visit and undergo surgical treatment. The representativeness in the SNUH cohort was also supported by the fact that our findings were validated in the nationwide cohort with strong statistical significance. International validation in East Asian countries would increase the generalizability.